# Compassionate Use of Encapsulated MKB-01 Fecal Microbiota Transplantation for Recurrent *Clostridioides difficile* Infection: A Single-Center Experience

**DOI:** 10.3390/microorganisms13092134

**Published:** 2025-09-12

**Authors:** Ángela Cano, Elisa Ruiz Arabi, Lourdes Ruiz, Borja José Nadales, Andrés Baumela, Manuel Recio, Isabel Machuca, Juan José Castón, Elena Pérez-Nadales, Julian Torre Cisneros

**Affiliations:** 1Instituto Maimónides de Investigación Biomédica de Córdoba (IMIBIC), 14005 Córdoba, Spain; angela.cano.sspa@juntadeandalucia.es (Á.C.); elisa.ruiz.arabi.sspa@juntadeandalucia.es (E.R.A.); lourdes.ruiz@imibic.org (L.R.); borja.nadales@imibic.org (B.J.N.); andres.baumela.sspa@juntadeandalucia.es (A.B.); isabelm.machuca@juntadeandalucia.es (I.M.); juanj.caston.sspa@juntadeandalucia.es (J.J.C.); elena.pereznadales@imibic.org (E.P.-N.); 2Servicio de Enfermedades Infecciosas, Hospital Universitario Reina Sofía, 14004 Córdoba, Spain; manuel.recio.sspa@juntadeandalucia.es; 3Centro de Investigación Biomédica en Red Enfermedades Infecciosas (CIBERINFEC), Instituto de Salud Carlos III, 28029 Madrid, Spain

**Keywords:** fecal microbiota transplantation, *Clostridioides difficile* infection, MKB-01, recurrence

## Abstract

Fecal microbiota transplantation (FMT) is a safe and effective treatment for recurrent *Clostridiodes difficile* infection (rCDI). However, experience with the oral biologic product MKB-01 remains limited. We describe a series of 13 patients with rCDI treated with FMT using MKB-01 capsules administered orally. Each patient received a single dose of 4 capsules (≥2.1–2.5 × 10^11^ microorganisms) with water after a 2 h fasting period. Antibiotic therapy was discontinued pre FMT. Clinical evaluation was performed at weeks 8 and 12. The mean number of prior recurrences was 1.5 (range: 1–3 episodes). In 12 patients (92.3%), FMT was administered after resolution of the current episode; in one patient (7%), it was administered on day 3 of fidaxomicin therapy, prior to symptom resolution. At week 8, clinical cure (Absence of baseline symptoms for at least 72 h) was achieved in 11 patients (84.6%). An additional patient (7%) responded to a second FMT. One recurrence occurred at 8 weeks and was resolved with a second FMT. Therefore, the overall clinical response rate after one or more FMTs was 12 out of 13 patients (92.3%). The procedure was well tolerated; only one patient experienced self-limited diarrhea. These findings support oral FMT with MKB-01 capsules as a safe and effective option for treating rCDI.

## 1. Introduction

*Clostridioides difficile* (formerly *Clostridium difficile*) is a microorganism etiologically associated with diarrhea driven by the overuse of broad-spectrum oral antibiotics, including those targeting anaerobes. Once primarily regarded as a nosocomial problem in developed countries, *C. difficile* infection (CDI) has become a global public health emergency over the past 15 years due to a significant increase in incidence. An epidemiological shift has also been observed, with CDI becoming an increasingly prevalent community-acquired infection. At the same time, recurrent cases and mortality have risen [1], attributable to the emergence of increasingly virulent strains, improved diagnostic capabilities, inappropriate use of antibiotics, and the progressive aging of the population [2]. In Spain, the incidence of nosocomial CDI is estimated at 10 episodes per 10,000 hospital stays, with recurrence rates ranging from 12% to 18% [3,4,5].

Over the past decade, there has been a notable increase in severe and fulminant forms of CDI. It is estimated that approximately 10% of hospitalized CDI patients experience therapeutic failure, colonic perforation (leading to peritonitis), or toxic megacolon [6].

First-line antibiotic treatments for CDI, such as vancomycin and fidaxomicin, are effective in controlling symptoms but paradoxically exacerbate dysbiosis. Vancomycin is non-selective and, besides targeting the vegetative forms of *C. difficile*, it also kills other Gram-positive members of the microbiota (particularly Firmicutes), thereby disrupting microbial balance and reducing the proportion of butyrate-producing (anti-inflammatory) strains. Although fidaxomicin is significantly more selective and inhibits sporulation—thus lowering recurrence rates—it still negatively affects the composition of the microbiome. Antibiotic treatment for each recurrence further disrupts microbiota diversity, progressively increasing the risk of subsequent recurrences, reaching 40–65% from the second episode onward [7]. This establishes a self-perpetuating cycle that can only be interrupted by restoring a normal intestinal microbiota.

Recurrent CDI (rCDI) and surgical intervention for fulminant forms represent not only a serious clinical problem but also substantial economic burdens. These costs are primarily due to prolonged hospital stays and the need for patient isolation, which limits the use of shared hospital rooms [8].

Rigorous data are required to confirm the benefits of fecal microbiota transplantation (FMT) in clinical scenarios characterized by dysbiosis [9]. FMT involves the infusion of stool from a healthy donor into the gastrointestinal tract of a patient with the aim of treating a specific disease associated with altered gut microbiota. It has proven to be an effective treatment for recurrent and refractory CDI. The efficacy of pharmacological treatments followed by FMT exceeds that of pharmacological treatments alone [9,10]. FMT is also a safe procedure with few adverse effects limited to mild and transient abdominal discomfort (such as bloating, flatulence, constipation, and nausea) [11]. Consequently, clinical management guidelines for CDI recommend FMT for rCDI [12,13,14]. Evidence also supports the efficacy of FMT in treating fulminant forms of CDI; however, its effectiveness is reduced in these cases, and patients often require repeated FMTs over short periods of time (sequential FMT protocols) [6].

These beneficial effects are due to the ability of FMT to restore gut microbiota diversity, which leads to several advantageous outcomes: (i) a healthy microbiota competes for intestinal niches and nutritional resources; (ii) it produces bacteriocins with bacteriostatic or bactericidal activity; (iii) it restores bile acid metabolism, thereby inhibiting the germination of *C. difficile* spores; (iv) it reestablishes intestinal barrier immunity [6]. Some experts are convinced that the sterile filtrate (which contains bacteriophges) plays a key role in curing CDI, while the bacteria-containing stool fraction primarily contributes to microbiome restoration and the prevention of recurrence [15,16,17,18]. MBK-01 is an investigational biologic drug consisting of lyophilized heterologous intestinal microbiota capsules (Full Spectrum & Purified Intestinal Microbiota, FSPIM).

The efficacy and safety of MBK-01, compared to fidaxomicin, have been previously evaluated in a randomized, open-label clinical trial (EudraCT 2020-004591-17; NCT 05201079) for both initial CDI episodes and recurrences. However, the results of this trial have not yet been published.

Following completion of this clinical trial, the AEMPS authorized compassionate use of MBK-01. Given the lack of published evidence, our objective is to evaluate the efficacy and safety of FMT using MBK-01 for the treatment of rCDI in a series of patients treated compassionately at our center.

## 2. Materials and Methods

### 2.1. Design

This study was approved by a del CEIm Provincial de Córdoba, with an approval letter dated 27 November 2024, and reference number SICEIA-2024-003053. This is an observational study involving medicinal products as defined in Article 2 of Royal Decree 957/2020 of November 3 regulating observational studies with medicinal products for human use in Spain. It is a descriptive study aimed at evaluating the efficacy and safety of MBK-01. The compassionate use program of MBK-01 for the treatment of rCDI via FMT was authorized by the AEMPS in 2023 following the completion of the phase III clinical trial MBK ICD-01 (EudraCT 2020-004591-17; NCT05201079). Since then, MBK-01 has been used in 13 patients at our center, who are the subjects of this study. Therefore, none of the patients included here participated in the aforementioned clinical trial.

### 2.2. Inclusion Criteria

Before offering patients MBK-01 under compassionate use, a benefit-risk assessment of the procedure was conducted. The treatment was offered only to male and female patients over 18 years of age with no controlled rCDI who had experienced at least two previous episodes of CDI, or a first severe episode with risk factors for recurrence. Eligible patients had either failed or were intolerant to approved pharmacological treatments for CDI.

An episode of CDI was defined as diarrhea with more than three liquid stools per day at onset with presence of *Clostridium difficile* toxin in stool. For detection of toxin, a screening was performed using a colored chromatographic immunoassay for the simultaneous qualitative detection of *Clostridium difficile* Glutamate Dehydrogenase (GDH), Toxin A and Toxin B in stool samples (CerTest *Clostridium difficile* GDH+Toxin A+B, CerTest Biotec S.L., Zaragoza, Spain). Positive stool samples were confirmed using a molecular method of detection of the *C. difficile* toxin B gene (tcdB) (BD MAX^TM^ Cdiff assay, Becton, Dickinson and Company, Franklin Lakes, NJ, USA).

### 2.3. Exclusion Criteria

To ensure patient safety, pregnant women and patients with a history of anaphylaxis to any food were excluded. Patients who had previously undergone FMT received antibiotics within the prior week, solid organ transplant recipients (except those transplanted over two years prior with good graft function), those with an absolute neutrophil count below 500 cells/μL, patients with swallowing difficulties, critically ill patients or those with an expected survival of less than 72 h, and patients unable to undergo proper follow-up were also excluded.

### 2.4. Procedure, Follow-Up, and Analysis

MBK-01 is an investigational biologic drug consisting of lyophilized heterologous intestinal microbiota capsules (Full Spectrum & Purified Intestinal Microbiota, FSPIM) manufactured by Mikrobiomik (Vizcaya, Spain). It is derived from multiple healthy donors formulated in gastro-resistant capsules with an enteric coating made of materials that allow delayed release of viable microbial content beyond the stomach, avoiding degradation in gastric acid and ensuring a targeted delivery [19]. This capsules undergo a rigorous clinical and microbiological evaluation process in accordance with the consensus guidelines established by the Spanish National Transplant Organization (ONT) and the Spanish Agency for Medicines and Health Products (AEMPS), the joint position statement issued in 2021 by the Catalan Society of Digestology and the Catalan Society of Infectious Diseases and Clinical Microbiology, as well as the applicable Spanish legislation, particularly Law 9/2014 of July 9, which regulates the donation, procurement, use, and coordination of substances of human origin [19,20]. Each dose consists of four oral capsules containing lyophilized intestinal microbiota (≥2.1–2.5 × 10^11^ microorganisms). The microbial composition is characterized by a median of 64.5% Bacillota (formerly Firmicutes), ranging from 37% to 91%, and Bacteroidota (formerly Bacteroidetes), which varies between 1% and 50%. Additionally, Actinomycota, Verrucomicrobiota, and Pseudomonota (formerly Proteobacteria) are consistently present across all donors, with different abundances (20). The microbial community diversity is assessed via metagenomic analysis (16 rRNA) ensuring that each donor harbors diverse, balanced microbiota representative of a healthy intestinal ecosystem [19]. MBK-01 is fully microbiologically characterized, formulated as an Investigational Medicinal Product (IMP), and lyophilized to ensure stability. Unlike other experimental formulations using fresh or frozen stool, MBK-01 is subject to comprehensive quality control and traceability [19,20].

Selected patients received four capsules of MBK-01 administered as a single dose with water. A fasting period of 2 h was required. Patients did not receive proton pump inhibitors or gastric motility stimulants. Treatment with oral fidaxomicin or vancomycin was permitted but discontinued before FMT (at least 24 h). The capsules were stored at 5 ± 3 °C following the manufacturer’s recommendations.

Patients were clinically evaluated at 8 weeks (clinical cure) and 12 weeks (clinical cure after repeated doses and recurrence) following FMT. Clinical cure was defined as the absence of baseline symptoms (diarrhea and others) for at least 72 h. Recurrence was defined as the presence of diarrhea lasting more than 48 h with detection of C. difficile toxin(s) after achieving clinical cure (Table 1).

Qualitative variables are expressed as number of cases (proportion) and quantitative variables as the mean (interquartile range). Since this is a descriptive analysis, no statistical tests have been used.

The chat-GPT tool has been used in order to improve the readability and understanding of the data included in Table 2 by standardizing their format.

## 3. Results

### 3.1. Representative Cases

Table 1 provides a description of the 13 patients included in the study. Several considered to be representative are presented below.

#### 3.1.1. Case 1

A 79-year-old male with chronic kidney disease on hemodialysis and recurrent urinary tract infections requiring antibiotic therapy presented with a CDI episode characterized by more than 20 bowel movements per day and general deterioration. Diarrhea initially responded to a 10-day course of oral vancomycin but recurred one week after completing treatment. The recurrence was treated with a course of fidaxomicin, resulting in clinical improvement. However, three weeks after completing fidaxomicin, diarrhea recurred, and an extended fidaxomicin regimen was initiated. On day 21 of treatment, FMT with MBK-01 was performed. The patient maintained clinical response at 8 weeks post-procedure.

#### 3.1.2. Case 2

A 45-year-old female undergoing chemotherapy for breast cancer developed CDI despite no prior antibiotic use. The patient presented with 10 daily episodes of liquid stools, hypovolemic shock, and impaired renal function. She received a 10-day course of fidaxomicin. Two weeks post-treatment, she experienced a recurrence of profuse diarrhea. An extended course of fidaxomicin was initiated, along with intravenous metronidazole for the first 10 days. FMT was performed on day 20 of treatment, resulting in complete clinical response at 8 weeks.

#### 3.1.3. Case 4

An 81-year-old male with chronic kidney disease secondary to diabetes and recurrent urinary tract infections developed CDI (eight bowel movements per day) without general deterioration after completing a course of quinolone therapy. He was treated with a 10-day course of oral vancomycin. One week after completing treatment, diarrhea recurred, and a 10-day course of fidaxomicin was initiated. The day after completing fidaxomicin, the patient received FMT with MBK-01. However, intermittent symptoms persisted, and a second FMT with MBK-01 was performed. Clinical cure was achieved and sustained at 8 weeks post-treatment.

#### 3.1.4. Case 6

A 49-year-old male on hemodialysis developed lymphoma and underwent chemotherapy. During the second chemotherapy cycle, he experienced a complicated urinary tract infection requiring treatment with ceftriaxone. One week later, he developed severe CDI with signs of sepsis. He was treated with a 10-day course of oral vancomycin. Although the septic parameters were resolved, diarrhea persisted, prompting a 10-day course of fidaxomicin, after which he became asymptomatic. However, one week after completing fidaxomicin, symptoms recurred, and a new 10-day course of oral vancomycin was initiated. At that point, the Infectious Diseases Department was consulted. The patient was stable and asymptomatic at the time. FMT with MBK-01 was performed following completion of the vancomycin course. After initial symptom control, diarrhea recurred at 8 weeks, and a second FMT with MBK-01 was administered, resulting in the sustained resolution of symptoms.

#### 3.1.5. Case 7

A 71-year-old male was admitted with healthcare-associated pneumonia. While receiving treatment with ceftriaxone and levofloxacin, he developed CDI with 10 liquid bowel movements per day. Symptoms resolved after a 10-day course of fidaxomicin, but diarrhea recurred shortly after completing treatment. Oral vancomycin was initiated but failed to control symptoms. After 3 days, treatment was switched to an extended fidaxomicin regimen, which was maintained for 21 days. Following initial symptom control, diarrhea reappeared one week after completing treatment. A new course of fidaxomicin was started, and FMT with MBK-01 was performed on day 10 of treatment. The patient maintained clinical response at 8 weeks post-FMT.

#### 3.1.6. Case 9

A 72-year-old female developed nosocomial pyelonephritis treated with piperacillin–tazobactam. She subsequently developed severe CDI with sepsis and was treated with a 10-day course of oral vancomycin. Diarrhea recurred shortly after, and a 10-day course of fidaxomicin was initiated. Following initial symptom control, a third recurrence occurred. Oral vancomycin was prescribed for 10 days, which was combined with intravenous metronidazole for the first 8 days. At the end of treatment, a single dose of bezlotoxumab was administered to prevent further recurrences. However, this preventive approach failed, and CDI recurred one week later. After two days of fidaxomicin treatment, FMT with MBK-01 was performed, resulting in stable clinical cure.

#### 3.1.7. Case 12

A 73-year-old female developed CDI in the postoperative period following a renal transplant. The patient received immunosuppression with tacrolimus, mycophenolic acid, and corticosteroids. CDI was treated with a 10-day course of oral vancomycin. One week after completing treatment, symptoms recurred. At this point, the Infectious Diseases Department was consulted, and a 10-day course of fidaxomicin was prescribed, followed by FMT with MBK-01 at the end of treatment. However, this therapeutic approach failed.

### 3.2. Description of the Grouped Cases

The grouped case data are provided in Table 2. All patients were male except one (92.3%). The mean age was 66.9 years (range, 39–89 years). Nine patients (69.2%) were older than 65. In 12 cases (92.3%), the initial CDI episode was associated with antibiotic treatment for an infection. One additional patient was undergoing active chemotherapy for breast cancer without prior antibiotic use.

Three patients (23.1%) were on hemodialysis. One patient (7.7%) was a renal transplant recipient receiving immunosuppressive therapy with tacrolimus, mycophenolic acid, and corticosteroids.

The mean number of previous recurrences before the current episode was 1.5 (range, 1–3 episodes). Of the nine patients (69.3%) with only one prior episode, four had been treated with fidaxomicin and five with oral vancomycin. One patient (7.7%) had two previous episodes, which were treated with oral vancomycin and fidaxomicin. Another patient with three prior episodes received a 10-day fidaxomicin course for the first episode, an extended fidaxomicin regimen for the second, and a combination of oral vancomycin, metronidazole, and bezlotoxumab for the third. The mean interval between the last recurrence and the episode during which FMT was administered was 1.4 weeks (range, 1–3 weeks).

In the current episode, FMT was administered once the episode was controlled and completed with fidaxomicin (7 patients, including 2 on extended regimens) or with vancomycin (5 patients, with 2 cases also receiving tigecycline). FMT was administered in only one case on the third day of fidaxomicin treatment, without prior control of diarrhea. This patient had three previous recurrences and had failed treatment with bezlotoxumab.

Clinical cure at week 8 was observed in 11 patients (84.6%). One additional patient was cured after a second FMT. Therefore, the overall clinical response rate after multiple FMTs was 12 cases (92.3%). The procedure was well tolerated. Only one patient experienced self-limited diarrhea.

## 4. Discussion

Currently, fidaxomicin is the recommended pharmacological treatment for the first episode of CDI, particularly in patients with risk factors for recurrence. Although its initial clinical efficacy is comparable to that of oral vancomycin (79–86%), the sustained clinical response of fidaxomicin is greater because it reduces recurrences. According to ESCMID, the 28-day recurrence rate is 26% with oral vancomycin compared to 15.9% with fidaxomicin [6]. However, this benefit decreases in rCDI [6]. The introduction of bezlotoxumab led to an additional 13–19% reduction in the risk of recurrence [21,22]. Unfortunately, this monoclonal antibody has been withdrawn from the market.

In 2021, three medical societies, the American College of Gastroenterology (ACG) [12], the Infectious Diseases Society of America (IDSA) [13], and the European Society of Clinical Microbiology and Infectious Diseases (ESCMID) [14], recommended FMT as a treatment for rCDI due to its proven efficacy and safety, particularly in severe cases with a high risk of recurrence. FMT can be performed via the upper or lower gastrointestinal tract using endoscopy, rectal enemas, tubes, or orally. Oral capsule administration is now widely accepted as highly effective, with efficacy comparable to that of colonoscopic administration [6]. FMT via oral capsules is a highly accessible procedure in routine clinical practice that has overcome many logistical challenges that previously limited its use, enabling even outpatient treatment and facilitating repeated multi-dose regimens. However, a major issue is the lack of standardization in encapsulated FMT products. Products vary in donor selection, preparation methods, production conditions (aerobic versus anaerobic), and storage (frozen versus lyophilized). Additionally, there is no consensus regarding the optimal administered dose, necessary gastrointestinal preparation, or concomitant use of medications [9].

A recent meta-analysis evaluated the efficacy and safety of oral FMT [9]. The pooled analysis (16 studies) showed an 85% clinical response rate to a single dose of FMT at 8 weeks post-procedure. When cases responding to repeated treatments were included, the clinical response increased to 93%. The safety analysis indicated that the procedure is safe. No related deaths were observed, and FMT produced only mild and self-limited gastrointestinal discomfort. Similarly, a previous meta-analysis of the efficacy of FMT via various routes of administration (oral capsules, endoscopic, and colonic) showed similar results of clinical cure at 8 weeks after the procedure, 84% in single FMT, and 94% with multiple FMT [23]. However, all published meta-analyses are limited by the small number of randomized clinical trials they analyze, although this is offset by the remarkable consistency of the results across studies [23,24].

The data appear to incontrovertibly demonstrate that the efficacy of FMT is independent of the route used to perform it. However, FMT via oral capsules does not require endoscopic procedures, can be administered at home, and can easily be repeated as needed. In our opinion, this should be the preferred route, particularly for elderly patients [25]. One limitation of some oral products is that they require many capsules, or the capsules are very large and difficult to swallow. This is not the case with MBK-01 as the FMT is administered with only 4 easy-to-swallow capsules.

Encapsulated preparations also differ in terms of their dosage, preparation, and storage conditions. There is also no agreement as to whether intestinal cleansing prior to FMT is necessary. Moreover, a recent meta-analysis found no evidence that these differences affect the efficacy of FMT in treating rCDI, although there is a trend suggesting that the greater the amount of microbiome administered, the greater the efficacy [9]. It is possible that such differences have a greater impact when FMT is used to treat other processes where it remains unclear whether dysbiosis is the cause or the consequence of the underlying process. Another recently published clinical trial [26] concluded that FMT therapy versus placebo did not reduce CDI recurrence. However, in some cases, the diagnosis of recurrence was not confirmed, and symptoms consistent with CDI were categorized as possible recurrence, which could be an important limitation of the study.

The results of our study using FMT with encapsulated MBK-01 show similar results to those of this meta-analysis with other oral products. Clinical cure at week 8 was 84.6% and increased to 92.3% when patients responding to two FMTs were included. It should be noted that all our patients were high risk. FMT was effective in one patient with 3 previous episodes who had recurrence after using bezlotoxumab. All but one of our patients received FMT after completing pharmacological treatment. The NCT05201079 trial [27] compared FMT with encapsulated MBK-01 versus fidaxomicin in the treatment of primary and recurrent episodes. The results have not yet been published but are available at ClinicalTrials.gov [27]. Clinical cure at week 8 was 81.25%. Notably, our patients received FMT after pharmacological treatment whereas participants in the clinical trial did not. This suggests that the efficacy of MBK-01 may be superior if administered to prevent recurrence after controlling an episode with pharmacological treatment. This needs to be tested in further studies. These results support having a marketed, readily available and standardized product to perform FMT.

Encapsulated MBK-01 is not the only biological product under study. Other laboratories such as Vedanta are developing similar products (VE303). VE303 is administered in a daily dose of one capsule for 14 days. The capsule contains 8 well-characterized commensal bacterial strains, which are grown in pure clonal cell banks. In the phase 2 clinical trial to prevent rCDI, FMT with VE303 reduced recurrences from 37% to 13.8% compared to placebo [28]. A phase 3 clinical trial (NCT06237452) is currently underway [28]. The results of all these studies will be available in the near future and offer a new perspective for the standardization of oral FMT.

The available evidence supports that FMT is a safe procedure, provide that strict protocols are followed to ensure that donors are healthy and free from potentially transmissible intestinal colonization [29,30]. In this regard, both the U.S. Food and Drug Administration (FDA) and expert groups have issued recommendations on the safety measures required when accepting fecal donations [31,32]. These recommendations should be followed by all stool banks. Products under development, including MBK-01, scrupulously comply with these recommendations. When implementing local donation and transfer procedures, which use locally obtained stools, it is more difficult to ensure that these procedures are followed. The future commercialization of products developed by external companies that meet these requirements will undoubtedly ensure that donation procedures are auditable and, therefore, safe.

Our study has important limitations, such as the small sample size, the lack of control group, and compassionate-use bias. Nevertheless our limited experience supports the efficacy and safety of oral FMT using MBK-01 and provides hypotheses for research based on MBK-01′s performance (e.g., potential use in post-antibiotic treatment as prophylaxis). The results obtained in the treatment of rCDI may open the possibility of exploring the efficacy of FMT in other diseases in which intestinal dysbiosis is present. These include liver cirrhosis with encephalopathy, inflammatory bowel diseases (especially ulcerative colitis), and intestinal colonization by multidrug-resistant bacteria, especially carbapenem-resistant Gram-negative bacilli. It is essential to conduct clinical trials to determine whether FMT can restore normal flora, resolve dysbiosis, and prevent infection and related mortality.

## 5. Conclusions

In conclusion, current scientific evidence supports the efficacy and safety of oral FMT using encapsulated microbiota for the treatment of rCDI, regardless of the procedures used in capsule preparation. Our limited clinical experience with MBK-01 indicates that it is a promising and easy-to-use product. However, there is an urgent need for the results of clinical trials to facilitate the integration of these products into routine clinical practice through formal commercialization. This will improve patient access to this therapeutic option in any center.

## Figures and Tables

**Table 1 microorganisms-13-02134-t001:** Clinical characteristics and efficacy outcomes in a cohort of 13 patients with recurrent *C. difficile* infection treated with encapsulated MKB-01 fecal microbiota transplantation.

Case	Age	Sex	Underlying Condition(s)	No. of Previous CDI Episodes	Treatments for Previous Episodes (Duration in Days)	Weeks Between Cure of Last Recurrence and Current Episode	Treatment for Current Episode in Addition to FMT (Duration in Days)	Day of FMT	Loose Stools at the Time of FMT (Number of Depositions)	Clinical Cure at 8 Weeks	Need for Repeated FMT (*n*)	Adverse Events
1	79	M	Chronic renal failure (hemodialysis), UTI	2	1st: Vancomycin (10) 2nd: Fidaxomicin (10)	3	Fidaxomicin (extended regimen)	21	3	Yes	No	None
2	45	F	Breast cancer (active chemotherapy)	1	Fidaxomicin (10)	2	Fidaxomicin (extended regimen) + metronidazole (10)	20	4	Yes	No	None
3	62	M	Hypogammaglobulinemia, chronic renal failure (hemodialysis), UTI	1	Vancomycin (10)	1	Fidaxomicin (10)	11	3	Yes	No	None
4	81	M	Diabetes, chronic renal failure, UTI	1	Vancomycin (10)	1	Fidaxomicin (10)	11	5	Yes	Yes (1)	None
5	89	M	Chronic renal failure, pneumonia	1	Fidaxomicin (10)	1	Fidaxomicin (10)	11	5	Yes	No	None
6	49	M	Chronic renal failure (hemodialysis), lymphoma (chemotherapy), UTI	2	1st: Vancomycin (10) 2nd: Fidaxomicin (10)	1	Vancomycin (10)	11	5	No	Yes (1)	None
7	66	M	Pneumonia	1	Vancomycin (10)	2	Fidaxomicin (10)	11	4	Yes	No	None
8	71	M	Pneumonia	2	1st: Fidaxomicin (10) 2nd: Vancomycin (5) + fidaxomicin (extended regimen)	1	Fidaxomicin (10)	11	3	Yes	No	None
9	72	M	UTI	3	1st: Vancomycin (10) 2nd: Fidaxomicin (10), 3rd: Vancomycin (10) + metronidazole (8) + bezlotoxumab (1 dose)	1	Fidaxomicin (2)	3	4	Yes	No	None
10	70	M	UTI	2	1st: Vancomycin (10), 2nd: Fidaxomicin (10) + metronidazole (8)	1	Oral and enema vancomycin (12) + tigecycline (12)	13	3	Yes	No	None
11	39	M	Recurrent prostatitis	1	Vancomycin (10)	1	Fidaxomicin (10)	12	3	Yes	No	None
12	73	M	Renal transplant, UTI	1	Vancomycin (10)	1	Fidaxomicin (10)	12	3	No	Not indicated	FMT failure
13	74	M	Diabetes, COPD, pneumonia	1	Fidaxomicin (10)	2	Oral and enema vancomycin (10) + tigecycline (8)	11	3	Yes	No	None

Note: Extended fidaxomicin regimen: 200 mg every 12 h for 5 days followed by 200 mg every 48 h from day 6 to 25. Abbreviations: CDI, *Clostridioides difficile* infection; COPD, chronic obstructive pulmonary disease; F, female; FMT, fecal microbiota transplantation; M, male; UTI, urinary tract infection.

**Table 2 microorganisms-13-02134-t002:** Summary of clinical characteristics.

Item	Response	N (%)	Mean (IQR)
Sex	Male	12 (92.3)	
	Female	1 (7.7)	
Age (years)	>65 years	9 (69.2)	66.9 (39–89)
First CDI episode related to antibiotics	Yes	12(92.3)	
	No (chemotherapy)	1 (7.7)	
Patients on hemodialysis	Cases	3 (23.1)	
Kidney transplant with immunosuppressants		1 (7.7)	
Prior recurrences			1.5 (1–3)
Prior recurrences treatment (1 episode)	Fidaxomicin	4 (30.8) *	
	Oral vancomycin	5 (38.5) *	
Prior recurrences treatment (2 episode)	Vancomycin + fidaxomicin	1 (7.7)	
Prior recurrences treatment (3 episode)	Vancomycin + metronidazole + bezlotoxumab	1 (7.7)	
Weeks between last recurrence and FMT			1.4 (1–3)
FMT administration timing	After fidaxomicin	7 (53.8)	
	After vancomycin (2 cases with tigecycline)	5 (38.5)	
	Early, during treatment	1 (7.7)	
Clinical cure at week 8	Cases	11 (84.6)	
Clinical cure after second transplant		1 (7.7)	
Overall clinical response after FMT		12 (92.3)	

* Percentage calculated based on patients with one previous episode (*n* = 9). Abbreviations: CDI, *Clostridioides difficile* infection; IQR, interquartile range.

## Data Availability

The original contributions presented in this study are included in the article. Further inquiries can be directed to the corresponding author.

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
