# Peer review of "Compassionate Use of Encapsulated MKB-01 Fecal Microbiota Transplantation for Recurrent *Clostridioides difficile* Infection: A Single-Center Experience"

_microorganisms, 2025, doi:10.3390/microorganisms13092134_

Round 1
Reviewer 1 Report
Comments and Suggestions for Authors
This is a valuable case series contributing real-world data on a novel encapsulated FMT product (MBK-01). However, the manuscript needs a major revision to enhance methodological clarity, reduce redundancy, improve the flow, and provide a more critical interpretation of results.
Decision: Major revision
Abstract
- The abstract provides a clear summary but lacks numerical detail for some aspects (e.g., duration between first and second dose).
- The sentence structure is occasionally awkward, e.g., “...was given on day 3 of fidaxomicin therapy without diarrhea control” would be clearer as “...was administered on day 3 of fidaxomicin therapy, prior to symptom resolution.”
- Consider clarifying what is meant by “clinical cure” in the abstract for standalone clarity.
Introduction
- The introduction is overly long and includes too much discussion (e.g., mechanisms of bacteriophage transmission and speculation on other diseases).
- Redundancy exists in parts explaining dysbiosis and CDI recurrence.
- The rationale for using MBK-01 specifically is introduced very late—move this up for better flow.
Materials and Methods
The methods lack sufficient detail about statistical analysis (if any), which is essential even in observational designs.
Results
- Overuse of detailed individual case reports in the main text makes it long and distracts from general trends. Move detailed case summaries to supplementary material and summarize key trends in main text.
- For table 1. Consider adding a column for "Adverse Events."
Discussion
- Lacks critical appraisal of limitations in their own study (small sample size, lack of control group, compassionate-use bias).
- Reduce repetitive content from introduction and earlier parts of the discussion.
- Provide hypotheses or future directions based on MBK-01’s performance (e.g., potential use post-antibiotic treatment as prophylaxis).
Conclusion
The current conclusion is overly concise and does not adequately reflect the study’s findings or implications. A single-sentence statement is insufficient for a scientific article. I recommend expanding this section to provide a more comprehensive and forward-looking summary.
Author Response
Comment 1: The abstract provides a clear summary but lacks numerical detail for some aspects (e.g., duration between first and second dose).
Response: Thank you for pointing this out. We agreed with this comment . This data have been added. On the other hand, there is no fixed interval between the first and second dose, as this depends on whether recurrence occurs and when it manifests. Therefore, the changes can be observed in lines 9 and 11 of the Abstract.
Comment 2: The sentence structure is occasionally awkward, e.g., “...was given on day 3 of fidaxomicin therapy without diarrhea control” would be clearer as “...was administered on day 3 of fidaxomicin therapy, prior to symptom resolution.”
Response: Thank you for pointing this out. We agreed with this comment . The sentence has been modified according with the reviewer´s suggestion. The changes can be observed in lines 9 and 10 of the Abstract.
Comment 3: Consider clarifying what is meant by “clinical cure” in the abstract for standalone clarity.
Response : Thank you for pointing this out. We agreed with this comment . It has been clarified according with the reviewer´s suggestion. The changes can be observed in line 10 of the Abstract.
Comment 4: The introduction is overly long and includes too much discussion (e.g., mechanisms of bacteriophage transmission and speculation on other diseases).
Response: Thank you for pointing this out. We agree that the introduction is long. We have reduced it avoiding redundancies. In our opinion, it describes the importance of CDI as clinical problem introducing the objective of our research: FMT with MBK-01. For this reason, it appears at the end of the introduction.
Comment 5: Redundancy exists in parts explaining dysbiosis and CDI recurrence.
Response: Thank you for pointing this out. We agreed with this comment. Redundancy has been eliminated.
Comment 6: The rationale for using MBK-01 specifically is introduced very late—move this up for better flow.
Response: Please see previous response.
Comment 7: The methods lack sufficient detail about statistical analysis (if any), which is essential even in observational designs.
Response: Thank you for pointing this out. We agreed with this comment. As this is a descriptive analysis, statistical tests were not used; however, we have added the method of presenting the variables. You can find the changes on page 5, section 2.4, paragraph 4.
Comment 8: Overuse of detailed individual case reports in the main text makes it long and distracts from general trends. Move detailed case summaries to supplementary material and summarize key trends in main text.
Response: Thank you for pointing this out. The journal's editorial guidelines require a minimum word count. If we follow the reviewer's suggestion, it would not be possible to meet this requirement. We leave the decision to the editor's decision.
Comment 9: For table 1. Consider adding a column for "Adverse Events."
Response: Thank you for pointing this out. This column has been added and you can find it in page 8.
Comment 10: Lacks critical appraisal of limitations in their own study (small sample size, lack of control group, compassionate-use bias).
Response : Thank you for pointing this out. Limitations of the study have been recognized. It can be found in the last paragraph of the discussion section.
Comment 11: Reduce repetitive content from introduction and earlier parts of the discussion.
Response : Thank you for pointing this out. Repetitive content has been eliminated and the discussion has been reduced.
Comment 12: Provide hypotheses or future directions based on MBK-01’s performance (e.g., potential use post-antibiotic treatment as prophylaxis).
Response : Thank you for pointing this out. It has been included and can be found in the second paragraph of the discussion section.
Comment 13: The current conclusion is overly concise and does not adequately reflect the study’s findings or implications. A single-sentence statement is insufficient for a scientific article. I recommend expanding this section to provide a more comprehensive and forward-looking summary.
Response : Thank you for pointing this out. The section has been expanded, moving it from discussion to conclusion.
Reviewer 2 Report
Comments and Suggestions for Authors
The study is a report on the compassionate use of encapsulated fecal microbiota MKB-01 transplantation for the treatment of recurrent Clostridium difficile infection.
There are some questions and amendments to consider:
- How were the MKB-01-encapsulated fecal microbiota obtained? How was their quality confirmed? And how do they differ from encapsulated fecal microbiota from healthy donors, which have been studied in numerous previous studies?
- The study concluded that FMT using oral MKB-01 capsules is an effective and safe treatment for rCDI. How did you reach this conclusion?
- Has Clostridium difficiletoxin testing (toxin B PCR) been performed? Details should be provided in the Materials and Methods section.
- What is the phyla distribution in the MKB-01 capsules and in the stool samples of the patients in the study? What are the similarities and differences between the capsules and the patients' stool samples?
- Has the diversity of the microbial community been assessed in MKB-01 capsules and in stool samples of patients participating in the study?
- In the Materials and Methods section, the authors elaborated on the patient cases in the study, but there are numerous evaluations that are not included, without which it is difficult to assess the efficacy of MKB-01 capsules. Therefore, the Materials and Methods section requires further evaluations and details, which should also be thoroughly discussed in the Discussion section.

Author Response
Comment 1: How were the MKB-01-encapsulated fecal microbiota obtained? How was their quality confirmed? And how do they differ from encapsulated fecal microbiota from healthy donors, which have been studied in numerous previous studies?
Response : Thank you for pointing this out. MBK-01 capsules are produced from fecal microbiota obtained from healthy donors who undergo a rigorous clinical and microbiological evaluation process. The screening process is extremely strict and complies with:
- The consensus between the Spanish National Transplant Organization (ONT) and the Spanish Agency for Medicines and Health Products (AEMPS).
- The joint position statement issued in 2021 by the Catalan Society of Gastroentherology and the Catalan Society of Infectious Diseases and Clinical Microbiology.
- The applicable Spanish legislation, particularly Law 9/2014, of July 9, regulating the donation, procurement, use, and coordination of substances of human origin.
The manufacturing process follows pharmaceutical quality standards under Good Manufacturing Practices (GMP). Stool donations are collected and processed under controlled anaerobic conditions, lyophilized, and then encapsulated. Each batch undergoes:
- Bacterial viability testing
- Pathogen absence testing (via culture and PCR)
- Endotoxin testing
- Sterility testing
- 16S rRNA sequencing for taxonomic profiling
Differences from other products: MBK-01 is fully microbiologically characterized, formulated as an Investigational Medicinal Product (IMP), and lyophilized to ensure stability. Unlike other experimental formulations using fresh or frozen stool, MBK-01 is subject to comprehensive quality control, traceability, and a standardized, reproducible route of administration.
It has been included in material and methods.
Comment 2: The study concluded that FMT using oral MKB-01 capsules is an effective and safe treatment for rCDI. How did you reach this conclusion?
Response : This conclusion is based in the results : « Clinical cure at week 8 was observed in 11 patients (84.6%). One additional patient was cured after a second FMT. Therefore, the overall clinical response rate after multiple FMTs was 12 cases (92.3%). The procedure was well tolerated. Only one patient experienced self-limited diarrhea ».
Comment 3: Has Clostridium difficile toxin testing (toxin B PCR) been performed? Details should be provided in the Materials and Methods section.
Response : Thank you for pointing this out. It has been included in material and methods. This change can be found on page 3, section 2.2, second paragraph.
Comment 4: What is the phyla distribution in the MKB-01 capsules and in the stool samples of the patients in the study? What are the similarities and differences between the capsules and the patients' stool samples?
Response: The phyla distribution in both MKB-01 capsules and patients have not been studied in this retrospective cohort. We are planning to do it in the future.
Comment 5: ¿Has the diversity of the microbial community been assessed in MKB-01 capsules and in stool samples of patients participating in the study?
Response: The diversity of the microbial community in both MKB-01 capsules and patients have not been studied in this retrospective cohort. We are planning to do it in the future.
Comment 6: In the Materials and Methods section, the authors elaborated on the patient cases in the study, but there are numerous evaluations that are not included, without which it is difficult to assess the efficacy of MKB-01 capsules. Therefore, the Materials and Methods section requires further evaluations and details, which should also be thoroughly discussed in the Discussion section.
Response: Thank you very much for your recommendation. To avoid being tedious, we have described only a few cases that we consider representative (not all of them). However, all cases are included in a table that provides detailed clinical data. In addition, the text includes an global analysis of the variables, supported by a corresponding table. If the reviewer specifies which additional data are required, we would be happy to include them.
Reviewer 3 Report
Comments and Suggestions for Authors
The investigators present a novel capsule-based (MKB-01) fecal microbiota transplantation (FMT) procedure used in thirteen cases of severe, recurrent Clostridioides difficile infection (CDI). The method yielded promising results, particularly in patients who had not responded to initial or repeated FMT treatments. This capsule represents a real-life application of a completed but not yet evaluated Phase III clinical trial. The manuscript is clear, logically organized. The topic is important because the need for standardized, safe, and easily administered fecal product.
While the results of this well-executed and thoroughly documented study are clearly positive, I would like to address several comments and suggestions for clarification prior publication.
Lines 53–54:
“… paradoxically exacerbate dysbiosis. This is because…”
The authors explain that vancomycin and fidaxomicin exacerbate dysbiosis because they do not eliminate spores. This is partially correct but incomplete. In fact, both vancomycin and fidaxomicin worsen dysbiosis, but the explanation given here is not fully convincing. Vancomycin is non-selective and, besides targeting the vegetative forms of C. difficile, it also kills other Gram-positive members of the microbiota (particularly Firmicutes), thereby disrupting microbial balance and reducing the proportion of butyrate-producing (anti-inflammatory) strains. Although fidaxomicin is significantly more selective and inhibits sporulation—thus lowering recurrence rates—it still negatively affects the composition of the microbiome. Recommendation: Please revise this section for microbiological accuracy and cite supporting literature.
Lines 94–95:
The claim is not only supported by the referenced study with a small sample size. In fact, the work of Ott et al. (DOI: 10.1053/j.gastro.2016.11.010), Varga et al. (DOI: 10.3389/fcimb.2023.1041384), and Sipos et al. (DOI: 10.3389/fcimb.2024.1424376) has demonstrated the efficacy of sterile, bacteria-free filtrates in treating CDI in studies with larger cohorts. I am convinced that the sterile filtrate plays a key role in curing CDI, while the bacteria-containing stool fraction primarily contributes to microbiome restoration and the prevention of recurrence. This view is further supported by the latter two studies.
Section 2.4: Procedure. Follow-up.
This section lacks a detailed description of the capsule used. What type of capsule was administered (gelatin, cellulose, or enterosolvent)? Why were patients not given a proton pump inhibitor (PPI) or prokinetics? How long does it take for the capsule to release its contents?
Furthermore, what was the clinical condition of the patients at the time of transplantation? How frequent were their diarrheal episodes? Was their CDI active at that point?
Upon reviewing the presented cases, it appears more accurate to state that the FMT successfully prevented recurrence rather than cured the patients, since they were receiving active anti-CDI treatment at the time. I kindly request clarification on this point.
Minor comments:
Lines 92–95: The bacteriophage hypothesis is promising, but evidence remains limited. Please moderate the tone and clarify its investigational status.
Table 1.: Consider adding a column indicating whether diarrhea was present at the time of FMT. This would help readers interpret treatment timing.
Recommendation:
The manuscript contributes a lot to the discussion around standardized oral FMT protocols. However, before publication I recommend major revisions with clarification of the issues noted above.
Author Response
Comment 1: Lines 53–54:
“… paradoxically exacerbate dysbiosis. This is because…”
The authors explain that vancomycin and fidaxomicin exacerbate dysbiosis because they do not eliminate spores. This is partially correct but incomplete. In fact, both vancomycin and fidaxomicin worsen dysbiosis, but the explanation given here is not fully convincing. Vancomycin is non-selective and, besides targeting the vegetative forms of C. difficile, it also kills other Gram-positive members of the microbiota (particularly Firmicutes), thereby disrupting microbial balance and reducing the proportion of butyrate-producing (anti-inflammatory) strains. Although fidaxomicin is significantly more selective and inhibits sporulation—thus lowering recurrence rates—it still negatively affects the composition of the microbiome. Recommendation: Please revise this section for microbiological accuracy and cite supporting literature.
Response : Many thanks for your valuable recommendation that has been incorporated to the text. This modification can be found on page 2, section 1, paragraph 3.
Comment 2: Lines 94–95:
The claim is not only supported by the referenced study with a small sample size. In fact, the work of Ott et al. (DOI: 10.1053/j.gastro.2016.11.010), Varga et al. (DOI: 10.3389/fcimb.2023.1041384), and Sipos et al. (DOI: 10.3389/fcimb.2024.1424376) has demonstrated the efficacy of sterile, bacteria-free filtrates in treating CDI in studies with larger cohorts. I am convinced that the sterile filtrate plays a key role in curing CDI, while the bacteria-containing stool fraction primarily contributes to microbiome restoration and the prevention of recurrence. This view is further supported by the latter two studies.
Response: Again, many thanks for your excellent recommendation that has been incorporated to the text. This modification can be found on page 4, section 2.4, paragraph 1.
Comment 3: Section 2.4: Procedure. Follow-up.
This section lacks a detailed description of the capsule used. What type of capsule was administered (gelatin, cellulose, or enterosolvent)? Why were patients not given a proton pump inhibitor (PPI) or prokinetics? How long does it take for the capsule to release its contents?
Response: The use of proton pump inhibitors (PPIs) or prokinetics is optional. The policy at our centre is not administer them. MBK-01 is formulated in gastro-resistant capsules with an enteric coating. Specifically, size 0 capsules are used, made of materials that allow delayed release of viable microbial content beyond the stomach, avoiding degradation in gastric acid.
According to the technical dossier:
- The capsules remain intact for at least 120 minutes at pH 1.2 (simulated gastric conditions)
- Desintegration occurs at pH > 5.5, once the capsules leave the stomach
- Estimated full release time is 30 to 60 minutes after gastric exit, depending on intestinal transit
This formulation ensures targeted delivery and protects bacterial viability. This modification can be found on page 4, section 2.4, paragraph 1.
Comment 4: Furthermore, what was the clinical condition of the patients at the time of transplantation? How frequent were their diarrheal episodes? Was their CDI active at that point?
Response: The patient was stable with respect their transplant status. Table 1 now includes the number of bowel movements for each patient at the time of FMT administration. It can be found on page 8.
Comment 5: Upon reviewing the presented cases, it appears more accurate to state that the FMT successfully prevented recurrence rather than cured the patients, since they were receiving active anti-CDI treatment at the time. I kindly request clarification on this point.
Response: Thank you for pointing this out. CDI was not controlled at the time of FMT. We understand prophylaxis as the administration of a medication when the patient is asymptomatic, with the aim of preventing recurrences. However, it is clear that the intended effect of treatment is not only to achieve clinical resolution of the episode but also to prevent future recurrences. We believe that this is adequately described in the text.
Comment 6: Lines 92–95: The bacteriophage hypothesis is promising, but evidence remains limited. Please moderate the tone and clarify its investigational status.
Response. The text has been modified according with this and previous recommendations.
Comment 7: Table 1.: Consider adding a column indicating whether diarrhea was present at the time of FMT. This would help readers interpret treatment timing.
Response: Thanks for your recommendation. It has been included now
Round 2
Reviewer 1 Report
Comments and Suggestions for Authors
The authors did reply to all comments and the manuscript could be accepted now.
Reviewer 2 Report
Comments and Suggestions for Authors
Thank you for revising the manuscript and providing clarifications. More details had to be added in the materials and methods section such as the microbial diversity determination methods in the transferred samples and identification the phyla within these samples. I hope this will be done in a future study, as stated in the authors' response.
Reviewer 3 Report
Comments and Suggestions for Authors
Thank you for your revisions. The revised manuscript is now suitable for publication.